# The importance of relativistic effects on two-photon absorption spectra in metal halide perovskites

Zimu Wei [1], Dengyang Guo[1], Jos Thieme[1], Claudine Katan [2], Valentina M. Caselli[1], Jacky Even [3]* & Tom J. Savenije [1]*

Despite intense research into the optoelectronic properties of metal halide perovskites (MHPs), sub-bandgap absorption in MHPs remains largely unexplored. Here we recorded two-photon absorption spectra of MHPs using the time-resolved microwave conductivity technique. A two-step upward trend is observed in the two-photon absorption spectrum for methylammonium lead iodide, and some analogues, which implies that the commonly used scaling law is not applicable to MHPs. This aspect is further confirmed by temperature-dependent conductivity measurements. Using an empirical multiband tight binding model, spectra for methylammonium lead iodide were calculated by integration over the entire Brillouin zone, showing compelling similarity with experimental results. We conclude that the second upward trend in the two-photon absorption spectrum originates from additional optical transitions to the heavy and light electron bands formed by the strong spin-orbit coupling. Hence, valuable insight can be obtained in the opto-electronic properties of MHPs by sub-bandgap spectroscopy, complemented by modelling.

---

[1] Department of Chemical Engineering, Delft University of Technology, 2629 HZ Delft, The Netherlands. [2] Univ Rennes, ENSCR, INSA Rennes, CNRS, ISCR (Institut des Sciences Chimiques de Rennes) - UMR, 6226 Rennes, France. [3] Univ Rennes, INSA Rennes, CNRS, Institut FOTON - UMR 6082, Rennes, France. *email: Jacky.Even@insa-rennes.fr; T.J.Savenije@tudelft.nl

Over the past few years, metal halide perovskites (MHPs) have attracted immense attention due to their extraordinary performance as the light absorbing layer in solar cells[1,2]. This exciting family of semiconductors exhibits large carrier mobilities[3,4], long charge carrier lifetimes[5,6], and a linear absorption coefficient over $10^5$ cm$^{-1}$ above the bandgap[1]. Recently, appreciable nonlinear absorption coefficients (and refractive indices) have been reported rendering these materials of interest for nonlinear photonics,[7] including two-photon-pumped lasers[8,9] and saturable absorption-based ultrafast pulsed lasers[10]. As a typical nonlinear process, two-photon absorption (2PA) features deep penetration depths and a quadratic dependence on the intensity providing opportunities for bio-imaging[11,12], photodynamic therapy[13], three-dimensional optical data storage,[14] microfabrication, testing the influence of environmental gases[15] and up-conversion lasing and amplification[16]. In addition, quantifying the wavelength dependence of the 2PA process is of fundamental interest, since the 2PA can yield detailed information of the energy-band structure in crystalline solids, which may not be accessible by ordinary optical absorption spectroscopy.

Transposing the initial concept proposed by M. Göppert[17] to a semiconductor, the simultaneous absorption of two photons can lead to an excitation of an electron from the valence band (VB) to conduction band (CB) via a virtual state. The generation rate of charge carriers, $n_0$ for the 2PA process is given by

$$\frac{dn_0}{dt} = \frac{\beta I^2}{2\hbar\omega} \qquad (1)$$

where $\beta$ (cm W$^{-1}$) is the 2PA coefficient, $I$ (W cm$^{-2}$) is the light intensity entering the sample and $\hbar\omega$ (J) is the incident photon energy. Since the absorption coefficient of 2PA is typically low, high intensities are required.

Despite the importance of revealing the 2PA spectrum[18], only a few studies have been devoted to the 2PA properties in MHPs performed by the Z-scan or optimized Z-scan technique yielding $\beta$ differing by several orders of magnitude between different MHPs or even between the same perovskites[19–21]. Furthermore, all Z-scan based measurements were performed at a single fixed wavelength. Wavelength dependent characterization was achieved by photoluminescence excitation spectroscopy, however, only up to $\hbar\omega/E_g = 0.524$[22]. Three-photon absorption was also evidenced below the threshold for 2PA in CsPbBr$_3$[23].

In this work, we have recorded the 2PA spectrum of methylammonium lead iodide perovskite (CH$_3$NH$_3$PbI$_3$) polycrystalline thin films using the time-resolved microwave conductivity (TRMC) technique, as well as CH$_3$NH$_3$PbBr$_3$ thin films and single crystals, and CsPbI$_3$ thin films. From the photoconductance induced by a nanosecond laser pulse, the initial number of photogenerated charge carriers is obtained for wavelengths ranging between 0.49 and 1 times the bandgap energy $(0.49E_g < \hbar\omega < E_g)$. It has been postulated that the Z-scan technique is prone to overestimate the value of $\beta$[7], since free carrier absorption can lead to an additional reduction of the transmitted light[24]. Since with the TRMC technique excess charges are probed by microwaves instead of light, this problem is surmounted. A two-step upward trend for the 2PA spectrum is observed in MHPs, which is explained for CH$_3$NH$_3$PbI$_3$ by a combination of multiple bandgap transitions derived from a symmetry-based empirical tight-binding model: a primary bandgap of 1.58 eV and a secondary bandgap above 2.25 eV owing to the spin-orbit coupling induced band splitting. Apart from 2PA we identify sub-bandgap linear absorption (SLA) at photon energies close to the band edge. Furthermore, we investigated for CH$_3$NH$_3$PbI$_3$ the impact of the tetragonal-to-orthorhombic phase transition on the 2PA coefficient, $\beta$, by changing the temperature.

## Results

**Sub-bandgap absorption processes.** Thin films of CH$_3$NH$_3$PbI$_3$ (about 200 nm thickness) were spin-coated on quartz. The X-ray diffraction pattern (Supplementary Fig. 1a) of the CH$_3$NH$_3$PbI$_3$ film displays strong reflections for the <110> and <220> planes confirming the formation of highly crystalline CH$_3$NH$_3$PbI$_3$. The 1PA spectrum (Supplementary Fig. 2a) shows a cut-off wavelength of 785 nm indicating a bandgap energy of 1.58 eV. To probe 2PA in the CH$_3$NH$_3$PbI$_3$ film, the sample was measured by the time-resolved microwave conductivity (TRMC) technique[25,26] for a wide range of photon energies varying from 0.775 to 1.55 eV. The light intensity was attenuated by an array of neutral density filters yielding light intensities varying from $I_{NO}$: $2 \times 10^{11}$ to $2 \times 10^{15}$ photons cm$^{-2}$ per pulse.

The TRMC technique can be used to study the dynamics of photoinduced charge carriers in low conductive semiconductor materials, in an electrodeless way. The photoconductance, $\Delta G$, of the samples was deduced from the laser-induced change in absorbed microwave power, $\Delta P$, normalized by the incident power, $P$ according to

$$\frac{\Delta P(t)}{P} = -K\Delta G(t) \qquad (2)$$

where $K$ is the sensitivity factor. To compare the photoconductance traces recorded at different intensities and wavelengths, we normalized $\Delta G$ for the incident photon intensities, $I_{NO}$ yielding: $\frac{\Delta G}{e\beta_0 I_{N0}}$. Here, $e$ is the elementary charge and $\beta_0$ is a dimensionless constant of the microwave cell. In Fig. 1a traces for different laser wavelengths and intensities are shown. For all photon energies, $\Delta G$ increases rapidly on excitation, followed by a slow decay due to recombination or immobilization of charges in trap states[5]. Interestingly, for photon energies of 1.45 eV traces for all intensities overlap, while for energies of 1.3 eV we observe a gradual increase in signal size with intensity. At 1.42 eV an intermediate regime is visible. Next, we plotted the intensity normalized maximum photoconductance values $\frac{\Delta G_{max}}{e\beta_0 I_{N0}}$ versus the incident intensity $I_{NO}$ for three different photon energies in Fig. 1b. At 1.45 eV the values of $\frac{\Delta G_{max}}{e\beta_0 I_{N0}}$ are almost constant with intensity suggesting a first-order excitation process. This process is explained by the optically induced electronic transitions of electrons from the VB to sub-bandgap levels, or from the latter to the CB, as depicted in Fig. 1c and denoted by sub-bandgap linear absorption (SLA). On the contrary, at 1.3 eV a clear linear dependence between $\frac{\Delta G_{max}}{e\beta_0 I_{N0}}$ versus $I_{NO}$ is observed which implies that the conductance is proportional to $I_{N0}^2$ agreeing with the 2PA process. In Fig. 1d, the mechanism is illustrated showing the generation of a charge carrier pair on absorbing two photons. At 1.42 eV an intermediate regime is visible: at low intensities $(I_{N0} < 2 \times 10^{14}$ photons cm$^{-2}$ per pulse), $\frac{\Delta G_{max}}{e\beta_0 I_{N0}}$ is almost constant. On increasing $I_{NO}$ the signal gradually increases, demonstrating the transition from predominantly SLA to the 2PA process. The lowest detectable photon energy is found to be 0.8 eV in close agreement with the energy threshold of 0.79 eV for the 2PA process. In short, the optical absorption below the bandgap by the CH$_3$NH$_3$PbI$_3$ film is explained as follows: in the far below-bandgap regime (0.8–1.4 eV), photoinduced charge carriers are predominantly generated by the 2PA process; a transition regime is found between 1.4 eV and 1.45 eV, where both SLA and 2PA are contributing to the signal; In the near band-edge regime (1.45–1.55 eV) only SLA was detected due to much higher densities of sub-bandgap levels compared to that in the far below-bandgap regime. We realize that the transition regime might shift depending on the quality of the MHP film. However, differences

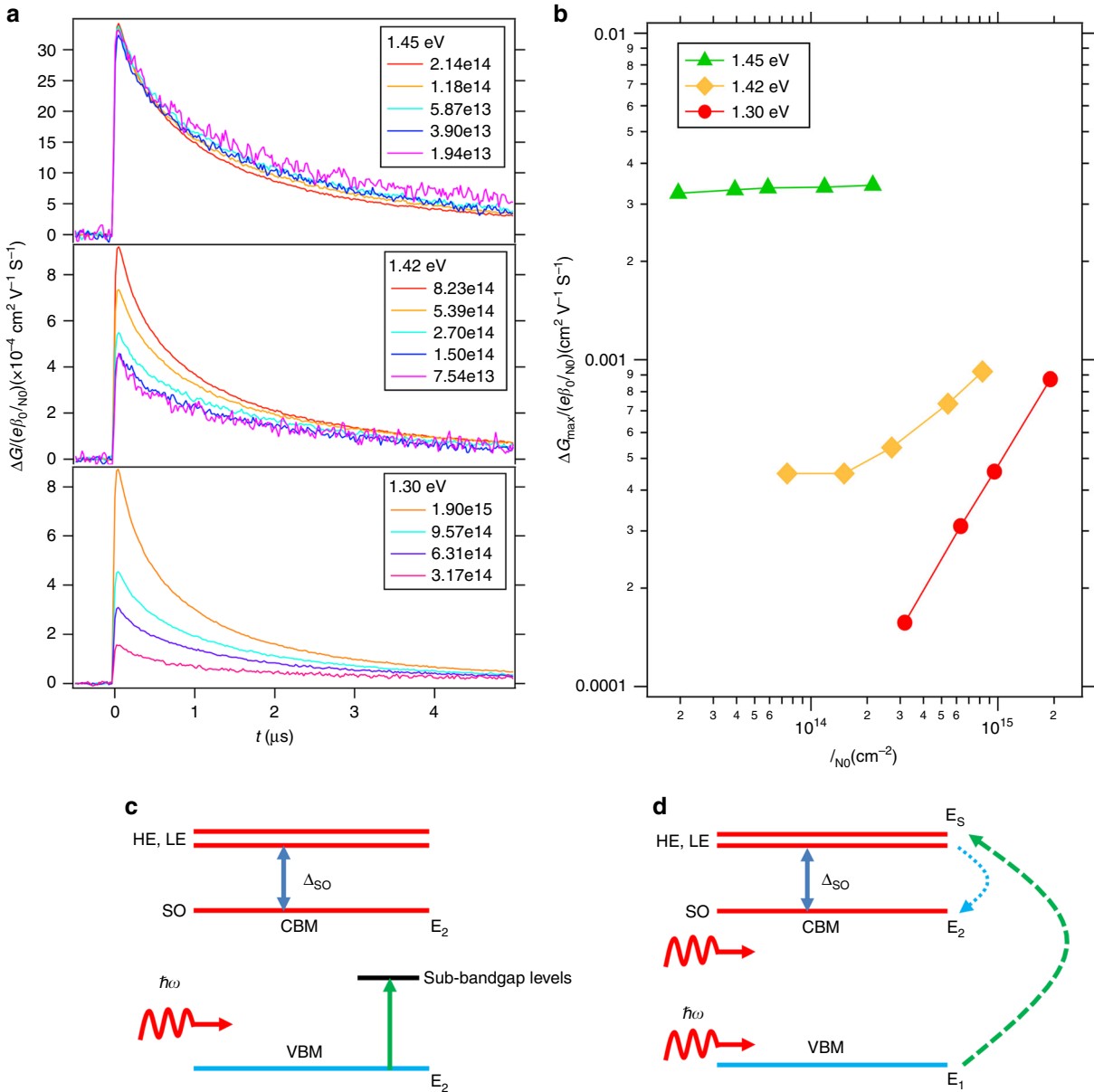

**Fig. 1** Charge carrier dynamics and absorption processes in sub-bandgap absorption. **a** $\frac{\Delta G}{e\beta_0 I_{N0}}$ as a function of time for a $CH_3NH_3PbI_3$ film measured at incident photon energies of 1.45 eV (top), 1.42 eV (middle) and 1.30 eV (lower). Legends show incident light intensity ($I_{NO}$) in photons cm$^{-2}$. **b** Corresponding $\frac{\Delta G_{max}}{e\beta_0 I_{NO}}$ versus $I_{NO}$. Schematics showing generation process of charge carriers by **c** optical excitation from or to sub-bandgap levels (sub-bandgap linear absorption (SLA)) and **d** by two-photon absorption (2PA). The initial ($E_1$) and final ($E_2$) states correspond to the valence band maximum (VBM) and the conduction band minimum (CBM). The intermediate states ($E_s$) have been chosen for illustration as the heavy and light electron states, separated from the spin-orbit split-off (SO) states by the spin-orbit coupling ($\Delta_{SO}$) interaction.

for this transition regime observed for five different $CH_3NH_3PbI_3$ films were less than 0.02 eV.

**Wavelength dependence of 2PA coefficient β.** To extract the 2PA coefficient $\beta$, the concentration of initially photogenerated charge carriers $n_0$ is first obtained from the maximum photo-conductance $\Delta G_{max}$ by

$$n_0 = \frac{\Delta G_{max}}{e\Sigma\mu\beta_0 L} \quad (3)$$

Where $\Sigma\mu$ is the sum of the electron and hole mobilities, $\beta_0$ is the dimensionless constant of the microwave cell, and $L$ is the sample thickness. As justified in Supplementary Fig. 3 and Supplementary Note 2, we assume that the initial signal is not affected by

recombination from which the number of photoinduced charge carriers, $n_0$ is deduced. Since $\Sigma\mu$ is an intrinsic property of the sample, we have deduced this value from a TRMC measurement above the bandgap (Supplementary Fig. 4)[26,27]. Next, we calculated the 2PA coefficient, $\beta$ according to

$$\beta = \frac{n_0 2\Delta t}{I_{N0}^2 \hbar\omega} \quad (4)$$

Here, $\Delta t$ is the full width at half-maximum (FWHM) of the laser pulse. To acquire the accurate laser pulse width, a sub nanosecond photodetector was used to record the pulse duration (Supplementary Fig. 5 and Supplementary Note 3). To obtain the actual intensity entering the sample, the light intensity measured by the power meter was corrected for reflection at the air/film interface

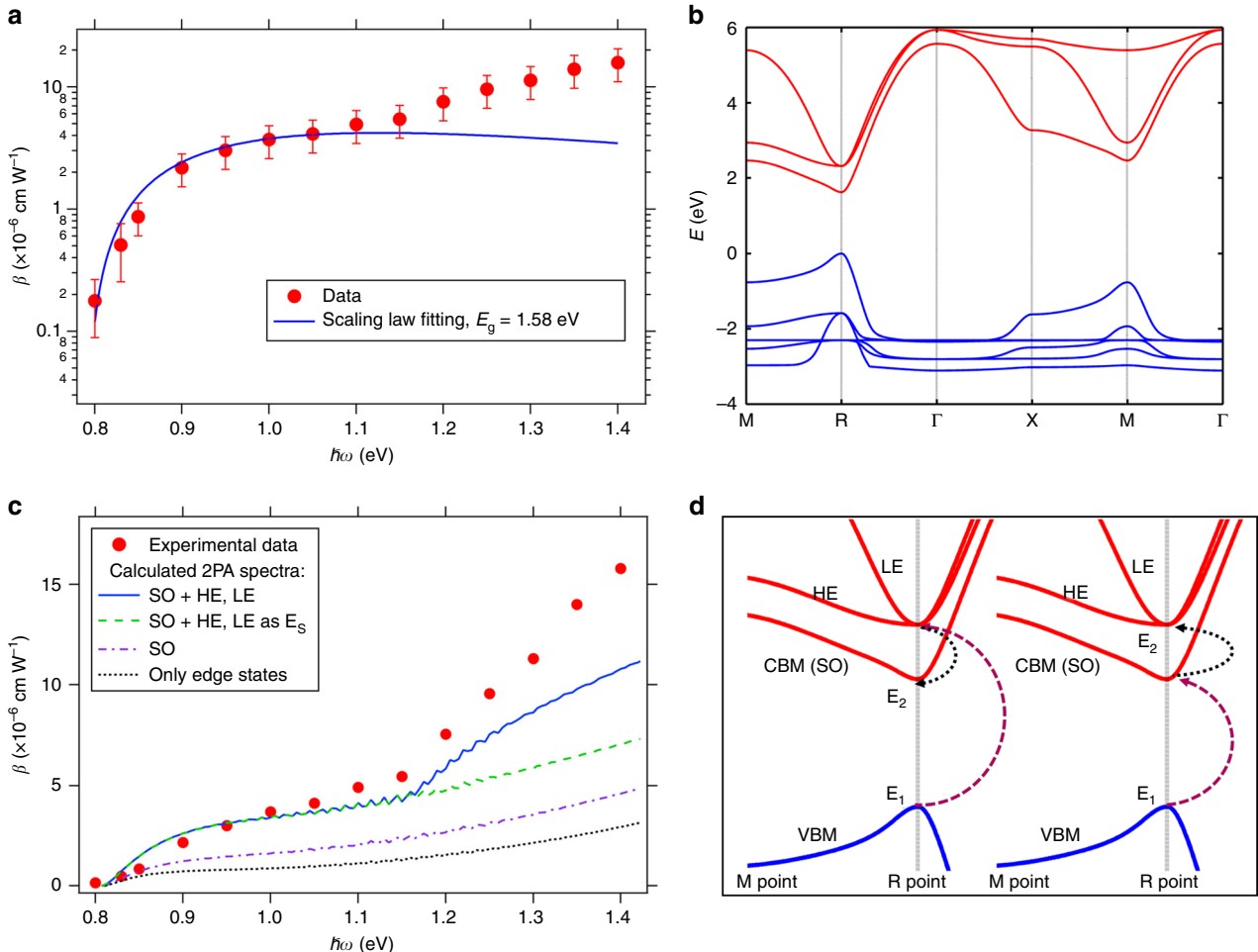

**Fig. 2** 2PA spectra and electronic band structure. **a** Experimental 2PA spectrum for $CH_3NH_3PbI_3$ compared to the scaling law, where 30 % error bars arise from sample variation. 50% error bars were introduced at 0.8 and 0.83 eV due to the limited data available. **b** Electronic band structure of cubic $CH_3NH_3PbI_3$ computed from an empirical tight-binding model. The figure is limited to 6 bands in the conduction band (red lines) and 10 bands in the valence band (blue lines). The bottom of the conduction band (CBM or SO bands) and the top of the valence bands (VBM) are both twice degenerated. The same model is used to compute the 2PA spectrum. **c** Full 2PA spectrum computed using the empirical tight-binding model (straight line). The dashed line represents the 2PA spectrum obtained by considering only the optical transitions to the bottom of the conduction band (spin-orbit split-off (SO) bands), but heavy (HE) and light (LE) electron states are still considered as virtual intermediate states ($E_s$). The dashed and dotted line corresponds to a computation where HE and LE bands are totally ignored. The dotted line represents the 2PA spectrum obtained by considering only the band edge states. **d** Some of the microscopic 2PA processes are schematically indicated by zooming the electronic band structure close to the bandgap. $E_1$ and $E_2$ are the initial and final electronic states, respectively. HE and LE are considered either as virtual intermediate or final states.

(Supplementary Fig. 6 and Supplementary Note 4). Basically, the values of $\beta$ (cm W$^{-1}$) can be derived from the slope of $\frac{\Delta G_{max}}{e\beta_0 I_{N0}}$ versus $I_{NO}$ in Fig. 1b.

Figure 2a displays $\beta$ as a function of photon energy, showing a rise by about two orders of magnitude on increasing photon energy. Considering that the reported $\beta$ for $CH_3NH_3PbX_3$ (X = Cl, Br, I) varies from $2.5 \times 10^{-4} (\hbar\omega/E_g = 0.524)$ to 272 cm MW$^{-1}$ ($\hbar\omega/E_g \sim 0.76$)[19,22] excluding the negative values[21], we can state that our values ranging from 0.18 ($\hbar\omega/E_g = 0.506$) to 15.8 cm MW$^{-1}$ ($\hbar\omega/E_g = 0.886$) are plausible. The most commonly used model for explaining the wavelength dependence of $\beta$ is the classical semiconductor scaling law[24], given by:

$$\beta = a\frac{1}{E_g^3}f(\hbar\omega/E_g) = a\frac{1}{E_g^3}\frac{\left(\frac{2\hbar\omega}{E_g}-1\right)^{\frac{3}{2}}}{\left(\frac{2\hbar\omega}{E_g}\right)^5} \quad (5)$$

Where $a$ (cm MW$^{-1}$) is a product of several constants including

the linear refractive index and material-independent constants. To further clarify the wavelength dependence of $\beta$, we first compare the wavelength-dependent function $f(\hbar\omega/E_g)$ given by the scaling law (Eq. 5) with our data. A bandgap energy, $E_g$ of 1.58 eV obtained from the 1PA absorption spectrum was used for the fitting. As shown by the blue curve in Fig. 2a, the large deviation mostly stems from the high energy range. The experimental wavelength dependence of $\beta$ shows a sharp rise at excitation energies close to 0.5 $E_g$ followed by a second upward trend. Although the band gaps of $CH_3NH_3PbBr_3$ and $CsPbI_3$ are different, basically similar trends are observed as shown in Supplementary Fig. 7 and Supplementary Note 5. The absorption is dominated for MHPs by SLA at energy higher than 0.9 $E_g$ thereby not being presented here. Interestingly, the second upward contribution at energy higher than 0.7 $E_g$ is opposite to the trend predicted by the scaling law, which predicts that $\beta$ should reach a maximum at 0.7 $E_g$.

Obviously, the scaling law is not directly applicable to MHPs, which is not surprising as it was designed for direct zinc blende

semiconductors[28,29]. In fact, the scaling law is based on a parabolic three-band model located at the Γ-point ($\mathbf{k} = (0, 0, 0)$), which is comprised of a 2-fold VB and a single CB[24]. However, the electronic band structure close to the bandgap has been found to be fundamentally different for $CH_3NH_3PbI_3$, where in the absence of relativistic effects, the CB is degenerated instead of the VB[30]. The dramatic spin-orbit coupling that results from the heavy metal atom, leads to spin-orbit split-off (SO) bands at the bottom of the CB, with heavy (HE) and light electron (LE) states lying at higher energies[31] (Fig. 2d). Moreover, the lowest energetic transition occurs at the R-point ($\mathbf{k} = (1/2, 1/2, 1/2)$) for the cubic phase of MHPs[32].

Besides, apart from a discussion on the matrix elements[29], the validity of the downward trend of the classical scaling law as $\hbar\omega/E_g$ approaching 1 (Fig. 2a) has been rarely addressed in the literature. It is therefore imperative to introduce a model suitable to reflect the main features of MHPs. Figure 2b shows the electronic band structure computed from the empirical tight-binding model recently developed for the cubic phase of MHPs[33]. This empirical model was designed to yield electronic band structures over the entire Brillouin zone, and not restricted to the proximity of the R-point. It compares well with more advanced first–principles calculations including many-body (e.g., sc-GW) corrections for interband electronic transitions up to about 2 eV above the electronic bandgap[33]. This model allows computing various properties at a low computational cost, similar to that of an effective mass model, but including the effect of spin-orbit coupling.

Here, we compute the 2PA coefficients of $CH_3NH_3PbI_3$ (see Supplementary Note 1 for details). Noteworthy, although the parabolic approximation usually holds for $\mathbf{k}$ values close to the high symmetry points, the non-parabolicity of the CB and VB has to be considered for photon energies away from the bandgap[34]. The empirical tight-binding model is including such non-parabolicity effects in a natural way. This non-parabolicity is expected to contribute to the wavelength dependence of $\beta$ for energy higher than $0.7 E_g$.

Consistently with our experimental data, the 2PA spectrum computed using the tight-binding model exhibits additional contributions related to the HE and LE bands (Fig. 2c, straight line). They may enter both the summations over the final states (index $c$ in Supplementary Equation 4) and over intermediate virtual $s$ states (index $s$ in Supplementary Equation 4 and Fig. 2d). Therefore, it is interesting to analyze in detail the various contributions. The dashed line represents in Fig. 2c the 2PA curve obtained by considering only the optical transitions to the bottom of the conduction band (SO bands), but HE and LE bands are still considered as possible intermediate $s$ states (Fig. 2d). By comparison to the straight line in Fig. 2c, we may therefore directly trace back the origin of the enhanced 2PA above 2.25 eV (Fig. 2a) to the presence of secondary gaps in the band structure induced by the spin-orbit coupling. Next, the dash-dotted line in Fig. 2c represents the 2PA curve obtained by removing also HE and LE bands from the list of possible intermediate $s$ states. By comparison to the dashed line, we conclude that HE and LE bands also influence the 2PA over the entire energy range as intermediate virtual states (Fig. 2d). This can be considered as a second, indirect, effect of the spin-orbit coupling. Finally, the dotted line represents a further restriction of the initial and intermediate states to the top of the valence bands. Comparison to the complete calculation (straight line) evidences that a computation of the 2PA restricted merely to band edge states is not justified. Thus, it is essential to consider also remote intermediate states. Noteworthy, it was impossible to include more bands away from the band edge states using our empirical tight-binding approach. This intrinsic limitation of the tight-

binding model, might be in principle overcome by switching to a completer and more accurate DFT calculation. However, the computational cost of such a 2PA DFT computation including many-body corrections to yield accurate band gaps as well as effective masses shall be extremely high and, to the best of our knowledge, has not yet been implemented in available computational software.

**Bandgap dependence of 2PA coefficient $\beta$.** In classical semiconductors, the dependence of the 2PA coefficient, $\beta$ as a function of the electronic bandgap has been discussed by considering various semiconductors with different band gaps[24]. Here, we might expect changes in $\beta$ in MHPs due to temperature-induced changes in the bandgap and the presence of temperature-related structural phase transitions (Supplementary Fig. 8)[34,35]. Hence, we investigated $\beta$ in $CH_3NH_3PbI_3$ by using a temperature-controlled microwave cell. The values of $E_g$ at each temperature ware determined by the cut-off wavelength from the linear absorption spectra (Supplementary Fig. 9). In contrast to the TRMC results at room temperature, at lower temperatures the sub bandgap absorption process at 1.43 eV is dominated by the 2PA process (Supplementary Fig. 10).

As depicted in Fig. 3a, in the tetragonal phase $\beta$ at a given photon energy increases, as the bandgap energy becomes smaller with decreasing temperature. This is consistent with the reported smaller 2PA coefficients observed at wider bandgap energies in $CH_3NH_3PbI_3$ powder[36] as well as $CsPbBr_3$ quantum dots[37]. This is basically the same trend as observed in Fig. 2a. Moreover, after rescaling the $\beta$ coefficient for its variation as a function of $E_g$ and $\hbar\omega$ according to the scaling law (Eq. 5), we do not observe a constant value over the entire energy range (Fig. 3b). We attribute this to additional contributions related to the HE and LE bands, confirming again the importance of relativistic effects to the 2PA process. For orthorhombic $CH_3NH_3PbI_3$ (squares in Fig. 3a), $\beta$ is invariably smaller than the values obtained for the tetragonal phase, which can be associated to the wider bandgap in the orthorhombic phase. In addition, it has been reported that both the M-point and R-point electronic states are folded back to the Γ-point in the orthorhombic phase due to a reduction of the Brillouin zone volume[38,39]. Hence, we expect that the 2PA in the orthorhombic phase should correspond to the transitions at the Γ-point. Although it has been shown that the oscillator strengths (Kane energies) remain comparable in the tetragonal and orthorhombic phases[39], it is important to note that the matrix element (e.g. comparable to the Kane energy that enters the prefactor $a$ in Eq. 5), the density of states as well as the allowed transitions regarding the 2PA will be different in the orthorhombic phase[40]. Therefore, the smaller values of $\beta$ observed in the orthorhombic phase can be explained by a combination of the above-mentioned factors.

## Discussion

In summary using the TRMC technique, we record the 2PA spectra of different metal halide perovskites, which demonstrate a two-step upward trend suggesting a primary bandgap in correspondence with the 1PA bandgap and an additional secondary bandgap. For $CH_3NH_3PbI_3$, sub-bandgap linear absorption is found to be the dominant process for photon energies close to the bandgap edge ($\hbar\omega > 0.9 E_g$) due to high density of trap states. A purpose build tight-binding model is designed to rationalize experimental 2PA spectra for $CH_3NH_3PbI_3$, which allows accounting empirically for spin-obit coupling and dispersion non-parabolicity over the entire Brillouin zone. Computed $\beta$ values are in good agreement with experimental values over the entire investigated spectral region. It reveals that the additional

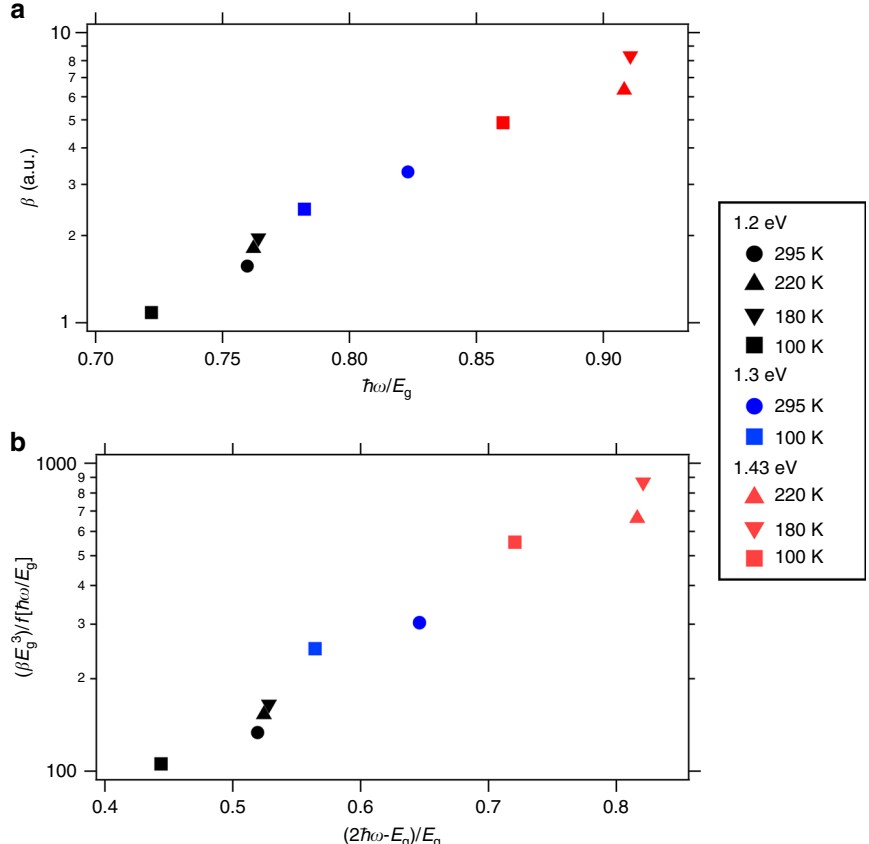

**Fig. 3** Bandgap dependence of 2PA coefficient $\beta$. **a** $\beta$ of $CH_3NH_3PbI_3$ obtained using temperature-dependent TRMC at different photon energies and temperatures. The values of $\beta$ calculated from the temperature-dependent TRMC measurements are expressed in arbitrary units due to the near infrared sensitivity of the temperature-dependent TRMC cell. However, this does not affect the trend for $\beta$. **b** Scaled 2PA coefficient $\beta$ versus normalized photon energies with respect to the material band gaps at different temperatures. Scaling has been performed according to the empirical scaling law (expression (22) of ref. [24]). Legend denotes the excitation energy and temperature for each data point. All corresponding data are summarized in Supplementary Table 1.

contribution to the 2PA starting at 2.25 eV for tetragonal $CH_3NH_3PbI_3$ can be attributed to a secondary bandgap at the R-point. This is traced back to the HE and LE bands that are separated from the SO conduction band edge states at the R-point as a result of relativistic effects. It is further demonstrated that they also contribute indirectly to the 2PA response over the entire energy range as intermediate virtual states. A negative correlation between the values of $\beta$ and the 1PA bandgap associated with temperature and structural phase is found. Bandgap dependent results confirm that the scaling law is inapplicable to the 2PA in MHPs. Overall, the simultaneous implementation of a 2PA experiment and a tight-binding model shows great promises to gain in-depth into the band structure of MHPs.

## Methods

**Preparation of the $CH_3NH_3PbI_3$ Films**. 13.93 mL $CH_3NH_2$ (4.24 g, 0.137 mol, 40% in methanol) was mixed with 15 mL HI (14.52 g, 0.114 mol, 57 wt% in water) in a 250 mL round-bottom flask. The system was immersed in the ice-water bath for 4 h with stirring. Subsequently, the precipitate was recovered by evaporation at 55 °C for 1 h. To purify the product, $CH_3NH_3I$ was washed by diethyl ether for three times, and dried at 60 °C in a vacuum oven for 24 h. $CH_3NH_3PbI_3$ thin films were prepared by the one-step method. Quartz substrates were washed with soap and de-ionized water, and sequentially soaked in acetone and ethanol with ultrasonic cleaning for 15 min, followed by a 10 min oxygen plasma cleaning. 477 mg $CH_3NH_3PbI_3$ (0.003 mol) and 379 mg lead acetate trihydrate ($Pb(CH_3COO)_2$. $3H_2O$) (0.001 mol) were dissolved in 1.54 mL DMF with the concentration of 37 wt % and stirred for over 30 min. Subsequently, 120 μL precursor solution was dropped on a cleaned substrate and then spin-coated at 2000 rpm for 45 s in the nitrogen atmosphere. After the substrate was dried at room temperature for 15 min

and annealed at 100 °C for 5 min, the samples are stored in a nitrogen-filled glove box before further characterization.

**TRMC measurements**. Nanosecond tunable laser pulses were produced from an integrated optical parametric oscillator (OPO) system (EKSPLA NT342 B-SH/ SFG). All samples measured in the near infrared (NIR) regime were always first measured at 500 nm to obtain referential TRMC results for estimating $\Sigma\mu$. Two extra filters (630 and 665 nm) were employed to avoid the interference of the visible light during the NIR measurements. After the measurements, the sample was measured at 500 nm again under preferably the same intensities as before to evaluate the effect of NIR irradiation on the sample. Only the samples with stable performance were further used for data analysis. All TRMC traces were averaged over at least 200 times to minimize the inaccuracy induced by the powermeter and the disturbance in the laser pulse. For temperature-dependent TRMC measurements, the experiments were carried out using another setup equipped with a liquid nitrogen cryostat. To switch from the visible regime to the NIR. regime, the alignment in the system was adjusted because of the different generation principles of the laser pulse. Upon cooling down from the room temperature (RT), the sample was recorded at 220, 180, and 100 K. The temperature was maintained for about 10 min before the measurement to assure the equilibrium of the system.

**Optical characterization**. Absorption spectra of perovskite thin films were acquired in a PerkinElmer LAMBDA 1050 UV/Vis/NIR spectrometer embodying a 150 mm InGaAs integrating sphere. The thin film sample was placed in a holder between the input light and the integrating sphere to measure the fraction of transmitted light $F_T$ and was clamped by a center mount accessory under an angle of 15° inside the sphere to obtain the total fraction of reflected and transmitted light $F_R + F_T$. Similarly, the fraction of reflected light $F_R$ was detected by placing the sample behind the integrating sphere. A Labsphere's Spectralon Reflectance Standard was used for the calibration of the reflection measurement, which provides 100% reflection.

**Tight-binding modeling**. The tight-binding model is a flexible empirical and symmetry-based atomistic model, widely used for classical semiconductors and has been recently adapted to the MHPs in ref. [33]. This atomistic method aims at empirically describing the chemical bonding in halide perovskites and the electronic band structure over the entire Brillouin zone, while keeping the computational effort at the level of multiband effective mass approaches and allowing descriptions of nanostructures up to a few millions atoms[41]. Its standard limitations are the lack of long-range interactions, on-site optical matrix elements and descriptions of excitonic effects. For this last two features not yet implemented for MHPs, also rarely taken into account for conventional semiconductors[42], explicit representations of atomic orbitals would be required[43]. Such an extension of the model could possibly lead to a considerable increase of the computational cost, especially when targeting nonlinear optical responses. Compared to the original work[33], the current tight-binding parameters have only been slightly adapted to better match the energetic position of the contributions to the 2PA related to the transitions involving HE and LE conduction band states. In addition, second nearest neighbor overlap integrals have also been added for p-orbitals to better capture the dispersions of the valence band[33].

## Data availability

The data that support the findings of this study are available from the corresponding author upon reasonable request.

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

## Acknowledgements

D.G. acknowledges the CSC for funding, File No. 201504910812. Dr Eline Hutter is gratefully acknowledged for the research idea.

## Author contributions

T.J.S. and D.G. conceived the idea and supervised the project. Z.W. performed $CH_3NH_3PbI_3$ film fabrication, most of materials characterization, TRMC measurements and data analysis. J.T. recorded laser pulse signals. V.M.C. prepared $CH_3NH_3PbBr_3$ and $CsPbI_3$ films. C.K. and J.E. performed theoretical calculations. Z.W., T.J.S., C.K. and J.E. co-wrote the paper.

## Competing interests

The authors declare no competing interests.
