## [Peer Review File · Nature Communications]

Reviewers' comments:

Reviewer #1 (Remarks to the Author):

The manuscript entitled "Two-Photon Absorption in Metal Halide Perovskites: Importance of Relativistic Effects" presents the investigation of 2PA characteristics in the polycrystalline MHP thin film using the TRMC technique. I agree with the author's statement that "the sub-bandgap absorption in MHPs remains largely unexplored." Despite the novelty of this work, this manuscript is not well organized. Therefore, I cannot recommend publishing this manuscript in Nature Communications. Some comments are listed below.

(1) Basically, it is necessary to make the manuscript more easy-read, including the correction of the title of supporting information. It seems to simply separate the text into supporting information so as not to exceed the length of the contents.

(2) In Figure S1, why does the entire XRD peak appear at higher angle than the reference peak? The major XRD peaks of $\text{CH}_3\text{NH}_3\text{PbI}_3$ are known as 14.06° (110), 28.40° (220), and 31.78° (310), respectively.

(3) On page 5, the notation description is missing for the equation.

(4) On page 6, the author mentioned that the transition regime might shift depending on the quality of the MHP film, but is there any reference for that? How many samples were used to measure? I think it is better to show error bars on the graph.

(5) On page 15, why is 2PA transition rate decreasing at wider bandgap energies? I understand that this result is consistent with the previous report, but I suggest that the author will give more explanations on this part.

(6) According to the introduction (page 5), I think another new aspect of this work is "we report, for the first time, the 2PA spectra including a temperature-dependence of MHP polycrystalline films." However, the results shown in this manuscript are insufficient to support this claim. Have you checked 2PA behavior at different photon energies depending on the temperature?

Reviewer #2 (Remarks to the Author):

In the manuscript entitled 'Two-Photon Absorption in Metal Halide Perovskites: Importance of Relativistic Effects' Z. Wei et al. report on sub band gap spectroscopy using the time-resolved microwave technique to record the two-photon absorption spectra of metal halide perovskite thin films. The wavelength dependent 2PA coefficient β is determined and a theoretical model is applied to compute β and compare with experimental results.

The manuscript is interesting and well written. The experiments appear to be well designed and performed.

However, I have concerns regarding the broader importance and implications of the work, which were not clear to me from the current manuscript text. For example, already the title (which is different from the one given for the SI) mentions 'Importance of relativistic effects'. It did not become clear to me what these 'relativistic effects' imply for the MHPs investigated or for their applications. In addition, from my impression the theory model and some of the description and statements in the text should be checked for relevance and the possibility to focus on the main message. Specifically, the authors state the limitations of their empirical tight binding approach by saying that it cannot be excluded that the reached agreement between experiment and model is 'a consequence of some fortuitous error cancelation'. Hence, what are the implications for the validity of the model?

Overall, although I find the described physics indeed interesting, I have to admit that I am not really convinced that the findings, as currently presented, justify publication in Nature Communications.

Some minor, technical comments:

- 1) Refs. 40, 41 are identical
- 2) Figure S4: β or $\beta_{>0}$?

Reviewer #3 (Remarks to the Author):

This is an interesting work about the quantification of two photon absorption (2PA) coefficient (beta in cm/MW) of lead halide perovskites by using time-resolved microwave conductivity (TRMC). The excitation energy dependence from $0.49E_g$ to E_g (the bandgap of perovskite) exhibits a two-step increase, which is rationalized by the spin-orbit split-off (SO) effect along with the heavy and light electron (HE and LE, respectively), namely a relativistic effect. The tight-binding model calculation supports the experimental results and the obtained 2PA coefficient is convincing. Therefore, I think this manuscript deserves publication in Nature Commun, almost as is.

Minor comments

1. The horizontal caption of Figures 2a and 2b should be $\hbar\omega$ (h-bar x omega).
2. The blue solid line and green dotted line in Figure 2b possibly correspond to the right and left panels in Figure 2d. Some labels (for example, i and ii) are recommended to add for an easy understanding.
3. For comparison, the cross section value in GM unit calculated from the beta value of 15.8 cm/MW ($\hbar\omega/E_g = 0.886$) is useful. Some literatures use the cross sectional values to quantify 2PA process. In particular, the perovskite family reportedly shows a very high cross section.
4. "is" should be included between "I (W/cm²)" and "the" and " $\hbar\omega$ (J)" and "the" on p. 3.

Reviewer #1 (Remarks to the Author):

The manuscript entitled “Two-Photon Absorption in Metal Halide Perovskites: Importance of Relativistic Effects” presents the investigation of 2PA characteristics in the polycrystalline MHP thin film using the TRMC technique. I agree with the author’s statement that “the sub-bandgap absorption in MHPs remains largely unexplored.” Despite the novelty of this work, this manuscript is not well organized. Therefore, I cannot recommend publishing this manuscript in Nature Communications. Some comments are listed below.

(1) Basically, it is necessary to make the manuscript more easy-read, including the correction of the title of supporting information. It seems to simply separate the text into supporting information so as not to exceed the length of the contents.

We thank the reviewer for his comment and agree that some parts might be less suited for a broad audience. Therefore, we tried to improve the clarity of our work in several instances, with major changes as follows:

- Figure 1 contains now additional panels showing the original traces. Corresponding text has been adapted.
- The way we express the photoconductance as given by the expression on page 5 is now fully explained. (see also comment 3)
- In order to clarify the description of our results, details of the tight binding modeling have been rigorously shifted to the supplementary information.
- Figure 3 and related discussion have been substantially modified.

(2) In Figure S1, why does the entire XRD peak appear at higher angle than the reference peak? The major XRD peaks of $\text{CH}_3\text{NH}_3\text{PbI}_3$ are known as 14.06° (110), 28.40° (220), and 31.78° (310), respectively.

The perovskite sample was characterized by an XRD machine using a Co X-ray source ($\lambda = 1.79\text{\AA}$). By recalculating the 2θ values for a Cu $K\alpha$ radiation source with $\lambda = 1.54\text{\AA}$, the major XRD peaks are indeed 14.13° , 28.52° and 32.01° in agreement with the values given by the reviewer. A note to this difference is included in the caption of Figure S1. For completeness we included also the XRD patterns of $\text{CH}_3\text{NH}_3\text{PbBr}_3$ and CsPbI_3 .

(3) On page 5, the notation description is missing for the equation.

- The way we express the photoconductance as given by the expression on page 5 is now fully explained.

(4) On page 6, the author mentioned that the transition regime might shift depending on the quality of the MHP film, but is there any reference for that? How many samples were used to measure? I think it is better to show error bars on the graph.

With this sentence we intended to state that the intensity at which we see the transition from linear to supra linear is expected to change with the density of sub band gap states, which is linked to the sample quality. The reason for including this remark is that for $\text{CH}_3\text{NH}_3\text{PbBr}_3$ this threshold intensity occurs at much higher intensities than for $\text{CH}_3\text{NH}_3\text{PbI}_3$. We adapted

this on page 6. The values found for the for the double photon absorption coefficients for samples of the different batches are within 30%.

(5) On page 15, why is 2PA transition rate decreasing at wider bandgap energies? I understand that this result is consistent with the previous report, but I suggest that the author will give more explanations on this part.

Although this is not the main topic of this paper we would like to thank the reviewer for giving us the opportunity to deepen our analysis. Firstly, we collected some more β values at different temperatures and photon energies. Then we replotted Figure 3 showing now 2 panels: Figure 3a: β vs excitation energy normalized to the bandgap energy; And Figure 3b: deviation from the empirical scaling law. A Table has been added to the SI. From the 2PA spectrum shown in Figure 3a obtained by tuning E_g (via changing the temperature) a similar upward trend as the spectrum at room temperature (Figure 2a) can be observed. We attribute this to additional contributions related to the HE and LE bands at the excitation energies where the 2PA process dominates. From Figure 3B it is evident that the β coefficient corrected for its variation as a function of E_g and $\hbar\omega$ according to the scaling law (**Equation 5**), does not yield a constant value over the entire energy range, again confirming the importance of relativistic effects to the 2PA process. Note that for the orthorhombic phase several elements, including DOS, matrix elements and allowed transitions affect the 2PA spectrum. These aspects are now discussed on p15.

(6) According to the introduction (page 5), I think another new aspect of this work is “we report, for the first time, the 2PA spectra including a temperature-dependence of MHP polycrystalline films.” However, the results shown in this manuscript are insufficient to support this claim. Have you checked 2PA behavior at different photon energies depending on the temperature?

We would like to thank the reviewer for this recommendation. In order to strengthen our analysis on the temperature dependence of the 2PA, we have performed additional measurements: Besides the temperature dependent β values at 1.2eV, we provide additional temperature dependent values at 1.3 eV and 1.43 eV. These new results also show smaller β values in orthorhombic phase (see also answer to comment 5). Unfortunately, due to technical limitations of the setup no more additional data points can be measured. The claim at the end of the introduction is now more specific.

Reviewer #2 (Remarks to the Author):

In the manuscript entitled ‘Two-Photon Absorption in Metal Halide Perovskites: Importance of Relativistic Effects’ Z. Wei et al. report on sub band gap spectroscopy using the time-resolved microwave technique to record the two-photon absorption spectra of metal halide perovskite thin films. The wavelength dependent 2PA coefficient β is determined and a theoretical model is applied to compute β and compare with experimental results.

The manuscript is interesting and well written. The experiments appear to be well designed and performed.

However, I have concerns regarding the broader importance and implications of the work, which were not clear to me from the current manuscript text. For example, already the title (which is different from the one given for the SI) mentions ‘Importance of relativistic effects’. We would like to thank the reviewer for giving us the opportunity to clarify and better describe the goals of the present combined experimental/structural study. We believe that our work more generally shows that classic solid-state physics concepts may be used in the future to exploit the potential of MHPs for non-linear effects. In recent years, there have been very few examples of new classes of semiconductor materials that may complement the well-known list of ‘conventional semiconductors’ for which the physical properties (such as the 2PA (ref 24)) have been analyzed over the past decades using solid state physics concepts and semi-empirical modeling. This was the first goal of the present study. The second goal of the study is to show that specific details of the electronic band structure (MHPs are unusual semiconductors) such as the spin-orbit coupling or the presence of temperature-induced structural phase transitions, have very specific effects on the 2PA spectra. Hence by revealing the 2PA spectra, specific knowledge about the band structure of MHPs including relativistic effects can be obtained. For that purpose, we adapted the Title, parts of the Abstract and Conclusions.

It did not become clear to me what these ‘relativistic effects’ imply for the MHPs investigated or for their applications. In addition, from my impression the theory model and some of the description and statements in the text should be checked for relevance and the possibility to focus on the main message. Specifically, the authors state the limitations of their empirical tight binding approach by saying that it cannot be excluded that the reached agreement between experiment and model is ‘a consequence of some fortuitous error cancelation’. Hence, what are the implications for the validity of the model?

We agree that this statement was misleading and did not bring much insight into the results of our modelling (figure 2). The theoretical evaluation of the 2PA coefficient is very satisfying up to the secondary band gap at about 1.2eV, which is in our opinion already a strong result of the simple tight binding approach implemented in this work. This approach indeed correctly describes non-parabolic dispersions. It leads also to the correct prediction of a secondary band gap in the 2PA. The remaining computational task would be to cure the underestimation of the 2PA above the secondary band gap (figure 2c) by including remote bands in the model as additional intermediate states. Besides, we have further modified the text so as to better highlight the effect of the band splitting induced by the giant spin orbit coupling on the 2PA coefficient (effect on linear optical properties has already been discussed by some of us, see for instance <https://doi.org/10.1021/jz401532q> and <https://doi.org/10.1021/jp503337a>).

Hence, we removed this statement and adapted the text discussing the above issues on page 13.

Overall, although I find the described physics indeed interesting, I have to admit that I am not really convinced that the findings, as currently presented, justify publication in Nature Communications.

Some minor, technical comments:

1) Refs. 40, 41 are identical

References are adapted

2) Figure S4: β or β_0 ?

Error has been corrected.

Reviewer #3 (Remarks to the Author):

This is an interesting work about the quantification of two photon absorption (2PA) coefficient (beta in cm/MW) of lead halide perovskites by using time-resolved microwave conductivity (TRMC). The excitation energy dependence from $0.49E_g$ to E_g (the bandgap of perovskite) exhibits a two-step increase, which is rationalized by the spin-orbit split-off (SO) effect along with the heavy and light electron (HE and LE, respectively), namely a relativistic effect. The tight-binding model calculation supports the experimental results and the obtained 2PA coefficient is convincing. Therefore, I think this manuscript deserves publication in Nature Commun, almost as is.

We thank the Reviewer for his detailed assessment of our work – we are delighted to receive such excellent in-depth review with useful comments.

Minor comments

1. The horizontal caption of Figures 2a and 2b should be $\hbar\omega$ (h-bar x omega).

Axis label is corrected.

2. The blue solid line and green dotted line in Figure 2b possibly correspond to the right and left panels in Figure 2d. Some labels (for example, i and ii) are recommended to add for an easy understanding.

The blue solid line includes both right and left panels. The green dotted line corresponds to the right panel. Color has been changed for an easy understanding.

3. For comparison, the cross section value in GM unit calculated from the beta value of 15.8 cm/MW ($\hbar\omega/E_g = 0.886$) is useful. Some literatures use the cross sectional values to quantify 2PA process. In particular, the perovskite family reportedly shows a very high cross section.

The usual relationship between two photon absorption cross-section σ_2 and two photon absorption coefficient β for chromophores in solution depends on the photon energy and also on the density of absorbing species (N) is: $\sigma_2(\hbar\omega) = \beta(\hbar\omega) \hbar\omega / N$. If we consider that the density of absorbing species in MAPbI₃ corresponds to 2 electrons (including spins) per unit cell, then $\sigma_2(\hbar\omega/E_g = 0.886) = 4.10^4$ GM. As in bulk semiconductors, usual practice is to report the TPA coefficient and not the cross section, we prefer not to convert to GM units in our manuscript.

4. “is” should be included between “I (W/cm²)” and “the” and “ $\hbar\omega$ (J)” and “the” on p. 3.

Is changed accordingly.

REVIEWERS' COMMENTS:

Reviewer #1 (Remarks to the Author):

Despite the answers to reviewer's comments, still authors do not provide the clear explanations. Therefore, I cannot recommend publishing this manuscript in Nature Communications.

Reviewer #2 (Remarks to the Author):

The authors have adequately addressed my comments. I can recommend publication.

Reviewer #3 (Remarks to the Author):

The authors appropriately answered my comments, and I believe that the revised manuscript is ready for acceptance.